# Driving factors in pediatric emergency department use: an ecological retrospective study

Denis Mongin[1,2]*, Hervé Spechbach[2,3], Joachim Marti[4], Frederic Ehrler[2,5], Johan N. Siebert[2,6]

1 Rheumatology division, University Hospitals of Geneva, Geneva, Switzerland, 2 Faculty of Medicine, University of Geneva, Geneva, Switzerland, 3 Primary Care Medicine Division, Department of Community and Primary Care Medicine, University Hospitals of Geneva, Geneva, Switzerland, 4 Department of Epidemiology and Health Systems, Center for Primary Care and Public Health (Unisanté), University of Lausanne, Lausanne, Switzerland, 5 Information Systems Directorate, University Hospitals of Geneva, Geneva, Switzerland, 6 Division of Pediatric Emergency Medicine, Department of Women, Children and Adolescents, University Hospitals of Geneva, Geneva, Switzerland

* Denis.Mongin@unige.ch

## Abstract

### Background

Pediatric emergency departments (PEDs) often face high volumes of low-acuity visits, reflecting gaps in primary care access and socio-economic disparities. We investigated how neighborhood socio-economic vulnerability, pediatrician availability, and proximity to the PED jointly influence PED utilization in Geneva, Switzerland.

### Methods

In this retrospective ecological study (Jan 2023-Dec 2024), we aggregated all PED visits for children aged 0–16 years by neighborhood and Canadian Triage Acuity Scale (CTAS) level. Neighborhood visit incidence (unique patients per child population) was modeled using mixed-effects regression against a composite socio-economic vulnerability index (NSVI), pediatrician density within a 2 km radius, and distance to the PED, incorporating an exponential decay function for distance and postal code as a random intercept.

### Results

There were 68,482 PED visits by 35,994 children (35.1% of Geneva under-16 population). Low-acuity visits (CTAS 4–5) comprised ~50% of encounters. Both distance and socio-economic vulnerability showed clear dose-response relationships, with stronger effects observed for lower-acuity visits, and no interaction effect between them. Overall, proximity accounted for up to 20.8% of non-urgent PED use, while neighborhood socio-economic vulnerability explained up to 19.7% of low acuity

**Data availability statement:** Deidentified data will be made available upon request, conditioned to the signature of a Data Transfer and Use Agreement and a Data Transfer and Processing Agreement with the University Hospital of Geneva, and the approval of the protocol by the Geneva Research Ethics Commission (https://submissions.swissethics.ch/en or ccer@etat.ge.ch, mentioning the concerned request number req-2025-00269).

**Funding:** The author(s) received no specific funding for this work.

**Competing interests:** The authors have declared that no competing interests exist.

visits across Geneva. Pediatrician density showed a modest inverse association for low-acuity visits only.

## Conclusions

Both proximity and socio-economic vulnerability are independent determinants of non-urgent PED use. Policies focusing only on primary care access risk missing key drivers of PED use, highlighting the need for locally tailored strategies such as community outreach near hospitals or programs to strengthen health literacy among families.

---

## 1. Background

Pediatric emergency departments (PEDs) are essential components of healthcare systems, ensuring 24/7 access to acute care for children. However, high utilization raises significant public health concerns, often reflecting gaps in primary care access, parental health literacy, and the relative convenience of emergency services [1–4]. In the United States, children under 15 account for a substantial share of emergency department (ED) visits, with over 26 million annual PED visits reported [5]. An estimated 54–87% of these visits are attributed to non-urgent conditions [6,7]. Similar trends have been documented across European healthcare systems [1,8], where increased PED attendance contributes to overcrowding, extended waiting times, operational inefficiencies, and a higher incidence of patients leaving without being seen [9,10]. These factors can lead to reduced patient satisfaction and a deterioration in the overall quality of emergency care.

A robust body of literature indicates that PED utilization is strongly influenced by socio-economic determinants. Children from lower socio-economic status (SES) families and those covered by public insurance consistently exhibit higher rates of ED use compared to their more affluent or privately insured counterparts [11]. In addition to socio-economic vulnerabilities, ethnicity and language also play influential roles in shaping patterns of PED use. Data from North America indicate that children from minority background are more likely to visit the ED, with disparities reflecting deeper systemic inequities such as poverty, limited health literacy, linguistic barriers, and the absence of a consistent medical home [11].

In addition to socio-economic determinants, access to primary care has also been proven to influence PED attendance [12]. Notably, studies have shown that enhancing access to high-quality primary care is associated with significant reductions in ED visits for low-acuity conditions [13].

Beyond individual socio-economic factors, the geographic and ecological context in which families live also significantly influences ED utilization. Geographic proximity to healthcare facilities is a well-established driver of care-seeking behavior [14,15]. Families living closer to an ED are more likely to use it, even for less urgent conditions [11]. Conversely, those in peripheral areas may delay care due to transportation or access barriers, often presenting with more severe illness.

While distance to PED, socio-economic conditions, and access to primary care are each recognized as key determinants of PED utilization, these factors are often interrelated and may interact in complex ways. Socio-economic disadvantage can limit access to primary care, just as geographic proximity to PED services can be shaped by underlying social and spatial inequalities. Moreover, regions with poor access to pediatricians often overlap with socio-economically deprived areas [16], further compounding barriers to appropriate care. Despite these interconnected dynamics, most existing studies have examined these determinants in isolation, without accounting for their combined or interactive effects. This fragmented approach may obscure important residual confounding, limiting our understanding of how these factors jointly influence PED utilization. Furthermore, comparing the relative impact of each of these factors is essential to identify the most influential drivers of PED use and to inform more targeted, effective interventions.

To address these gaps, this retrospective ecological study systematically examined the demographic, socio-economic, the density of pediatricians and spatial profiles of children presenting to a tertiary PED over a two-year period. Our aim was to analyze patterns of PED use in relation to the availability of primary care, socio-economic background, and geographic proximity to the PED, in order to better understand the relative contribution of these factors.

## 2. Methods

### 2.1. Study design

We conducted a retrospective ecological study of PED visits to Geneva's tertiary hospital over a two-year period. Geneva is divided into 476 neighborhoods (median population: 693; median area: 0.33 km²), grouped into 65 postal zones. Neighborhood-level PED utilization among children aged 0–16 was analyzed by triage acuity using the five-level Canadian Triage and Acuity Scale (CTAS: 1=resuscitation, 2=emergent, 3=urgent, 4=less urgent, 5=non-urgent) [17]. Due to low numbers, CTAS levels 4 and 5 were combined (hereafter referred to as level 4*). Neighborhoods with fewer than 10 children were excluded.

### 2.2. Setting

Geneva is a densely populated urban canton (530,246 residents, including 86,835 children under 16), with substantial cross-border commuter traffic [18]. The Geneva University Hospitals (HUG) is a 2,095-bed tertiary care institution comprising eight hospital sites and two clinics.

### 2.3. Spatial data

All spatial and population data used in the study are based on official population statistics and the delimitation of Geneva neighborhoods as provided openly by the Geneva State Geomatics Services [19].

### 2.4. Outcome

The primary outcome was the neighborhood-level incidence of PED use per triage level, defined as the number of unique pediatric patients per 100 children aged 0–16 in the neighborhood population. If a child had multiple visits at different triage levels, he counted one for each of the triage level concerned. To compute the primary outcome, all visits information from 01/01/2023 to 31/12/2024 were retrieved the 14/02/2025. Time of arrival, birthdate, address, and triage information were obtained. Patients' addresses were then geocoded using the state geocoding API [20], and PED incidence was calculated by dividing the number of visits in each neighborhood by the age-specific neighborhood population (provided by the Geneva State Geomatics Services).

### 2.5. Exposures of interest

The primary exposures were distance to the PED, neighborhood socio-economic status, and pediatrician density. Distance to PED was measured as the straight-line distance from the geographic centroid of each neighborhood to the PED.

Socio-economic Status was assessed using the Neighborhood Socio-economic Vulnerability Index (NSVI), a composite score developed by the Centre d'analyse territorial des inégalités à Genève (CATI-GE), a joint university–state institute dedicated to studying socio-economic inequalities in Geneva [21], The NSVI ranges from 0 to 6 and is constructed from state-provided indicators including income, unemployment, and social benefits (full criteria are provided in Supplementary Table 1 in S1 File). Although the NSVI has not undergone formal psychometric validation, it is used operationally by cantonal authorities to guide the allocation of targeted funding to deprived neighborhoods, supporting its face and content validity. The index has also demonstrated consistent associations with health outcomes in prior peer-reviewed research, including a study conducted by our group in which higher NSVI scores were associated with reduced access to COVID-19 testing and worse disease outcomes [22].

Pediatrician density was derived from the Swiss Federal Register of Medical Professions [23] from which the addresses of all non-hospital pediatricians practicing in the Canton of Geneva were extracted. Each address was geocoded using the state geocoding API [20] to obtain precise geographic coordinates. For each neighborhood, the number of non-hospital pediatricians located within a fixed 2 km radius of the neighborhood centroid was counted and expressed per 1,000 children. This radius-based measure reflects the availability of pediatric primary care from a family's perspective, as it captures the nearby supply of providers irrespective of administrative boundaries.

### 2.6. Statistical analysis

#### 2.6.1. Main analysis.
We modeled neighborhood PED incidence by acuity level with a nonlinear mixed-effects regression:

$$Y_{i,j} \sim A\exp\left(\frac{-\ln(2) \times dist_{i,j}}{\lambda}\right) + (b_0 + b_j) + c \times NSVI_{i,j} + d \times (PEDdensity_{i,j} - 1) + \epsilon_{i,j}$$

With $i$ indexing neighborhoods and $j$ indexing postal codes,

- $Y_{i,j}$: incidence of PED use in neighborhoods $i$ and postal code $j$

- $A$: Maximum distance-related increase in PED incidence

- $\lambda$: the halving distance (distance at which the distance effect reduces by half)

- $c$, $d$: Effects of NSVI and pediatrician density, respectively

- $b_0$: baseline PED use incidence for a wealthy neighborhood (NSVI = 0), pediatrician density of 1 per 1000 children, and zero distance effect.

- $b_j$: random intercept by postal code

Absence of spatial correlation in the regression residuals, and measure of the correlation at the postal code level, was assessed using Moran's Index [24]. Quality of the regression was assessed using the Efron pseudo R-squared [25], which ranges from 0 to 1, with 1 indicating a perfect fit. All spatial statistical analyses were conducted using R [26], *data.table* for data management, and *sp* and *sf* [27] for handling spatial data.

#### 2.6.2. Sensitivity analysis.
We stratified the main model by age groups (0–5 vs. 6–15 years) and time of visit, distinguishing between school hours (7 AM-6 PM), non-school hours (7 PM-6 AM), and weekends. Additionally, we tested for an interaction between the NSVI and distance using an extended model that included an interaction term for the nonlinear exponential effect of distance and NSVI (see Supplementary Material for full model specification in S1 File).

#### 2.6.3. Counterfactual analyses.
To quantify potential reductions in PED use, we simulated four scenarios using the adjusted model and Geneva: placing all neighbourhoods at a fixed distance of 5 km from the PED; setting all

neighbourhoods to the lowest NSVI (i.e., 0); assuming a uniform pediatrician density of 2 per 1,000 children; and combining all three conditions simultaneously.

**2.6.4. Ethical consideration.** The study was considered as quality-of-care study and was exempt from formal ethical review under the Swiss Human Research Act (HRA), as confirmed by the Geneva Research Ethics Commission (req-2025–00269), therefore waiving the requirement for informed consent. The use of personal data irrespective of the informed consent of the patient to compute the outcome was further approved by HUG's institutional board for non-HRA research.

## 3. Results

### 3.1. Demographics

Between January 1, 2023, and December 31, 2024, the PED recorded 68,482 visits (Table 1) by 35,994 unique patients, representing 35.1% of the pediatric population (Table 2). CTAS level 1 (highest acuity) accounted for 2.1% of visits, mainly for respiratory issues. Level 2 comprised 11.3%, mostly respiratory and infectious cases. Level 3 represented 36.2%, with common complaints including respiratory, gastrointestinal, and musculoskeletal conditions. Level 4* was the largest group (50.5%), primarily involving infectious, gastrointestinal, musculoskeletal, dermatologic, ear, nose and throat (ENT), and ophthalmologic presentations.

Higher acuity was associated with younger age (median 3.5 years at level 1 vs. > 5 years at level 4*) (Table 1), a higher proportion of male patients, fewer residents of Geneva (dropping from 88.9% at level 4* to 81.9% at level 1), and a greater proportion of Swiss nationals and French speakers (Table 2). Low-acuity visits peaked between 7 and 10 AM (up to 55%) and increased by 10 percentage points during the summer months. The age distribution showed an inverse U-shape, peaking at around 60% among children aged 4–8 and declining to 35% for both younger and older children (Supplementary Fig 1 in S1 File).

### 3.2. Neighborhoods aggregated statistics

No missing data were present in the aggregated data

**Table 1. Descriptive statistics and demographic characteristics of PED visits between January 1, 2023, to December 31, 2024 at Geneva University Hospitals, for all acuity triage levels (Overall), and per Canadian Triage Acuity Scale (CTAS) level (1: vital resuscitation, 2: emergent, 3: urgent, 4*: less urgent (4) and non-urgent (5)).**

|  | Overall | CTAS = 1 | CTAS = 2 | CTAS = 3 | CTAS = 4* |
|---|---|---|---|---|---|
| N visits | 68482 | 1428 | 7712 | 24732 | 34518 |
| Age (median [IQR]) | 4.98 [1.80, 9.93] | 3.49 [1.19, 8.00] | 3.75 [0.97, 9.94] | 5.20 [1.61, 10.89] | 5.12 [2.19, 9.37] |
| LWBS (%) | 1519 (2.2) | 0 (0.0) | 38 (0.5) | 221 (0.9) | 1252 (3.6) |
| Presenting complaint (%) |  |  |  |  |  |
| Respiratory | 11624 (17.0) | 1046 (76.5) | 2596 (33.8) | 4850 (19.7) | 3132 (9.1) |
| Gastrointestinal | 10941 (16.0) | 23 (1.7) | 622 (8.1) | 4922 (20.0) | 5374 (15.6) |
| Infectious | 9886 (14.5) | 25 (1.8) | 1264 (16.5) | 2545 (10.3) | 6052 (17.5) |
| Musculoskeletal | 9648 (14.1) | 8 (0.6) | 844 (11.0) | 3735 (15.1) | 5061 (14.7) |
| Dermatology | 7829 (11.5) | 3 (0.2) | 343 (4.5) | 1868 (7.6) | 5615 (16.3) |
| ENT – Ophthalmology | 7628 (11.2) | 62 (4.5) | 118 (1.5) | 1852 (7.5) | 5596 (16.2) |
| Neurological | 4673 (6.8) | 113 (8.3) | 357 (4.6) | 2445 (9.9) | 1758 (5.1) |
| Genitourinary | 1964 (2.9) | 3 (0.2) | 355 (4.6) | 855 (3.5) | 751 (2.2) |
| other | 4038 (5.9) | 85 (6.2) | 1183 (15.4) | 1598 (6.5) | 1172 (3.4) |

**Table 2.** Descriptive statistics and demographic characteristics of unique patients attending PED between January 1, 2023, to December 31, 2024 at Geneva University Hospitals, for all acuity triage levels (Overall), and per Canadian Triage Acuity Scale (CTAS) level (1: vital resuscitation, 2: emergent, 3: urgent, 4*: less urgent (4) and non-urgent (5)). A patient attending several times the PED during the period for various CTAS level is counted in each CTAS level. As a consequence, the sum of all CTAS levels is superior to the Overall column.

| | Overall | CTAS=1 | CTAS=2 | CTAS=3 | CTAS=4* |
|---|---|---|---|---|---|
| Number of patients | 35994 | 1244 | 6095 | 17405 | 22326 |
| Total incidence (% of population) | 35.1 | 1.2 | 5.9 | 17.2 | 22.4 |
| Sex (%) | | | | | |
| Male | 19413 (53.9) | 719 (57.8) | 3468 (56.9) | 9475 (54.4) | 11963 (53.6) |
| Female | 16576 (46.1) | 525 (42.2) | 2627 (43.1) | 7928 (45.6) | 10360 (46.4) |
| Indeterminded | 5 (0.0) | 0 (0.0) | 0 (0.0) | 2 (0.0) | 3 (0.0) |
| Living in Geneva state (%) | 30513 (84.8) | 1006 (80.9) | 5120 (84.0) | 14895 (85.6) | 19451 (87.1) |
| Primary language (%) | | | | | |
| French | 25370 (70.8) | 940 (75.9) | 4621 (76.2) | 12600 (72.7) | 15097 (67.8) |
| English | 2139 (6.0) | 66 (5.3) | 339 (5.6) | 960 (5.5) | 1306 (5.9) |
| Portuguese | 1694 (4.7) | 45 (3.6) | 221 (3.6) | 809 (4.7) | 1167 (5.2) |
| Spanish | 1545 (4.3) | 45 (3.6) | 213 (3.5) | 721 (4.2) | 1048 (4.7) |
| Albanian | 901 (2.5) | 26 (2.1) | 122 (2.0) | 416 (2.4) | 628 (2.8) |
| Ukrainian | 585 (1.6) | 8 (0.6) | 63 (1.0) | 231 (1.3) | 473 (2.1) |
| Arabic | 480 (1.3) | 18 (1.5) | 72 (1.2) | 220 (1.3) | 307 (1.4) |
| Italian | 408 (1.1) | 10 (0.8) | 67 (1.1) | 190 (1.1) | 259 (1.2) |
| Other | 2698 (7.5) | 80 (6.5) | 343 (5.7) | 1190 (6.9) | 1966 (8.8) |
| Nationality (%) | | | | | |
| Swiss | 16013 (44.6) | 580 (46.7) | 2971 (48.8) | 8143 (46.9) | 9298 (41.7) |
| French | 3724 (10.4) | 160 (12.9) | 712 (11.7) | 1794 (10.3) | 2136 (9.6) |
| Portuguese | 2131 (5.9) | 73 (5.9) | 360 (5.9) | 1050 (6.0) | 1409 (6.3) |
| Kosovar | 1160 (3.2) | 36 (2.9) | 163 (2.7) | 544 (3.1) | 808 (3.6) |
| Spanish | 1146 (3.2) | 39 (3.1) | 176 (2.9) | 534 (3.1) | 748 (3.4) |
| Italian | 1060 (3.0) | 36 (2.9) | 165 (2.7) | 507 (2.9) | 657 (2.9) |
| Ukrainian | 775 (2.2) | 11 (0.9) | 76 (1.2) | 311 (1.8) | 625 (2.8) |
| Brazilian | 713 (2.0) | 12 (1.0) | 101 (1.7) | 317 (1.8) | 505 (2.3) |
| Other | 9184 (25.6) | 296 (23.8) | 1359 (22.3) | 4171 (24.0) | 6094 (27.4) |

**3.2.1. 1 Pediatrician and socio-economic indicators.** A total of 120 private pediatricians operated across 67 practices, yielding a crude density of 1.6 per 1,000 children (vs. 1.2 nationally) [27,28]. Pediatrician density varied substantially across the neighborhoods, with a median of 1.32 [0.80, 1.97] pediatricians per 1000 children. The study included 357 neighborhoods, with a median pediatric population of 183 [IQR 79–330] and a median distance of 3.84 km [IQR 2.29–6.88] from the PED. NSVI scores showed 50.7% of neighborhoods had no vulnerability, while 5.3% met all six NSVI indicators, qualifying as highly vulnerable. Distributions of key indicators are visualized in Supplementary Fig 2 in S1 File.

**3.2.2. 2 PED statistics.** The incidence of PED use varied substantially across neighborhoods (Fig 1), with a mean neighborhood-level incidence of 33.9% (SD 16.0%) across all triage levels.

In the multivariable analysis, significant spatial dependence of PED utilization was observed for CTAS levels 2, 3, and 4*, with the strength of this dependence increasing as triage acuity decreased (Table 3). For CTAS 2, the mean neighborhood-level incidence was 5.8%, with a distance-related variation of 4.8 percentage points [95% CI: 2.9–6.7], representing 82%

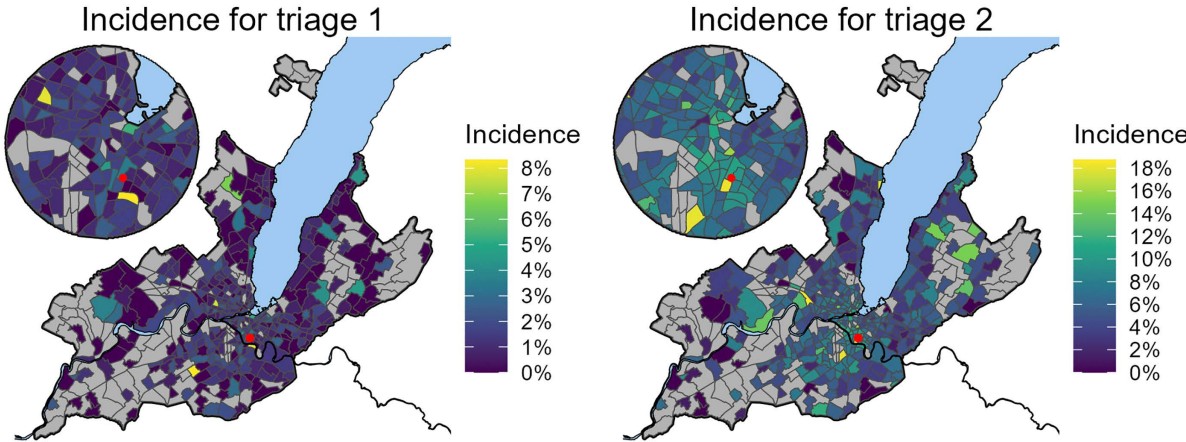

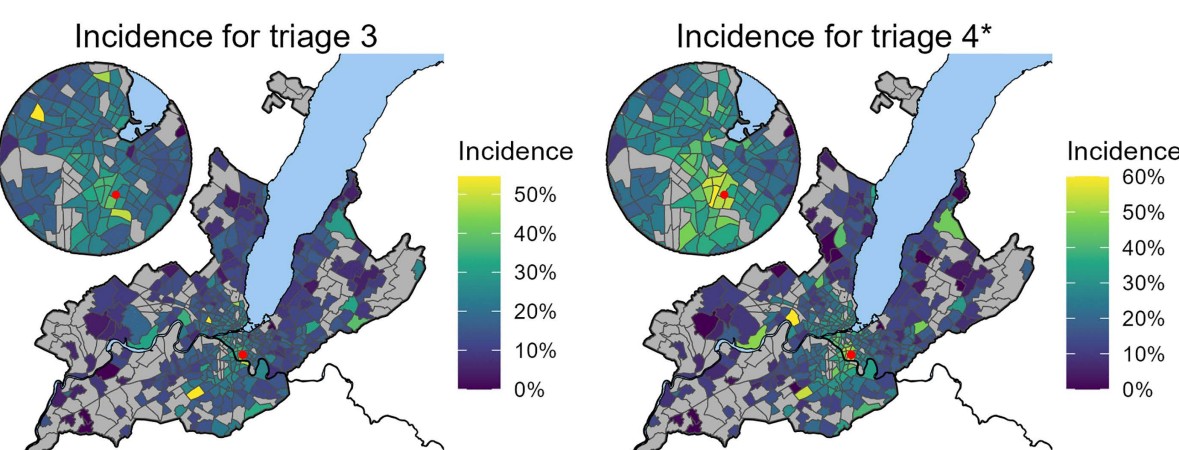

**Fig 1. Spatial distribution of PED incidence.** Incidence of PED use during 2023-2024 (Number of children going to PED during the period as a percentage of the number of children in the neighborhood), per neighbourhood and Canadian Triage Acuity Scale (CTAS) level (2: emergent, 3: urgent, 4*: less urgent (4) and non-urgent (5)). The Red point is the location of the PED in Geneva. Delimitations of the rivers and lake has been obtained from Open Street Map.

of the mean incidence. The spatial effect decreased by half approximately every 0.85 km [95% CI: 0.31–1.4]. For CTAS 3, the distance-related variation was 14 percentage points [95% CI: 10–18] (83% of the mean incidence), with a halving distance of 2.8 km [95% CI: 0.74–4.8]. For CTAS 4*, the variation was even larger, at 27 percentage points [95% CI: 22–31], representing 130% of the mean incidence, but with a faster decay over distance, halving every 1.7 km [95% CI: 1.1–2.3]. The spatial extent of the distance effect was similar for CTAS 3 and 4* (approximately 5 km) but the amplitude of the effect was markedly greater for CTAS 4* (Fig 2).

Socio-economic vulnerability was also significantly associated with PED use for CTAS 2–4*, with the effect size increasing as acuity decreased. Each additional NSVI point increased CTAS 2 incidence by 0.43 percentage points [95% CI: 0.26–0.6], totaling a 2.56-point gap (44% of the mean incidence) between least and most vulnerable areas. For CTAS 3 and 4*, these gaps were 8.4 (52%) and 13.2 (64.8%) percentage points, respectively.

**Table 3. Results of the multivariable regressions modeling the incidence of PED use during 2023-2024 as a function of distance to the PED, neighborhood socio-economic vulnerability, and pediatrician density, stratified by Canadian Triage Acuity Scale (CTAS) level (1: vital resuscitation, 2: emergent, 3: urgent, 4*: less urgent (4) and non-urgent (5)). *A* and λ represent the amplitude and characteristic length of the exponential spatial decay, respectively. *c* is the coefficient for pediatrician density (expressed as the number of pediatricians per 1000 children), *d* is the coefficient for Neighbourhood Socio-Economic Vulnerability Index (NSVI), and b denotes the baseline incidence. The percentage in parenthesis represents the marginal variation induced by the variable compared to the mean incidence.**

| CTAS coeff | 1 | 2 | 3 | 4* |
|---|---|---|---|---|
| Spatial effects: maximum amplitude (A) | 0.9 [95% CI: 0.4; 1.4]*** (83.6%) | 4.8 [95% CI: 2.9; 6.7]*** (82.6%) | 14 [95% CI: 10; 18]*** (83.6%) | 27 [95% CI: 22; 31]*** (129.9%) |
| Halving distance (λ, in km) | 1.9 [95% CI: −0.63; 4.5] | 0.85 [95% CI: 0.31; 1.4]** | 2.8 [95% CI: 0.74; 4.8]** | 1.7 [95% CI: 1.1; 2.3]*** |
| pediatrician density (c) | −0.075 [95% CI: −0.18; 0.031] (−7%) | −0.1 [95% CI: −0.39; 0.18] (−1.7%) | −0.76 [95% CI: −1.4; −0.068]* (−4.6%) | −1.4 [95% CI: −2.2; −0.53]** (−6.7%) |
| socio-economic effect (d) | 0.054 [95% CI: −0.012; 0.12] (30%) | 0.43 [95% CI: 0.26; 0.6]*** (44.4%) | 1.4 [95% CI: 1.1; 1.8]*** (52.2%) | 2.2 [95% CI: 1.8; 2.7]*** (64.8%) |
| baseline incidence (b) | 0.07 [95% CI: −0.97; 1.1] (6.6%) | 3.6 [95% CI: 1; 6.2]** (62.5%) | 2 [95% CI: −5.7; 9.6] (12.1%) | −2.1 [95% CI: −10; 6.2] (−10.3%) |
| Mean incidence (% of children) | 1.1 | 5.8 | 16 | 21 |
| Pseudo $R^2$ (%) | 0.053 | 0.24 | 0.53 | 0.63 |
| Moran Correlation index of the residuals | −0.066 | 0.044 | 0.028 | 0.0078 |
| Moran correlation index of the random effect coefficients (postal code level) | 0.15 | 0.11 | 0.23*** | 0.16* |

Higher pediatrician density was significantly associated with lower PED use for CTAS 3 and 4* only. Each additional pediatrician per 1,000 children reduced incidence by 0.7 [95% CI: 0.07–1.4] and 1.4 [95% CI: 1.1–1.8] percentage points, respectively, corresponding to 4.6% and 6.7% of the mean incidence. Residual spatial autocorrelation (Moran's I) was below 0.1 and non-significant in all models, while spatial autocorrelation in the postal-code random intercepts was significant for CTAS 3 and 4* but below 0.3 (0.23 and 0.16 respectively). Model fit (Efron's pseudo $R^2$) improved as acuity decreased, reaching a maximum at 0.63 for CTAS 4* (Table 3).

### 3.3. Sensitivity analysis

No interaction was found between distance and NSVI, indicating that the effect of distance was consistent across socio-economic levels (Supplementary Table 1 in S1 File). Subgroup analyses showed that both distance and socio-economic effects were more pronounced among older children (≥6 year) than younger children (≤5 years) (Supplementary Tables 2 and 3; Supplementary Figs 3 and 4 in S1 File). Similarly, all effects were stronger during non-school hours and weekends compared to school hours (Supplementary Tables 4-6; Supplementary Figs 5-7 in S1 File.

### 3.4. Counterfactual approach

Model-based estimates predicted 980, 5,291, 15,231, and 19,474 unique patients per year for CTAS levels 1–4*, respectively, closely matching the observed data (Table 4 vs. Table 1). Counterfactual scenarios estimated reductions in visits under three conditions: 1) setting all neighborhoods at an equal distance of 5 km was predicted to reduce visits by 571 patients (−10.8%) for CTAS 2, 1,829 (−12%) for CTAS 3, and 4,050 (−20.8%) for CTAS 4*; 2) assuming the least

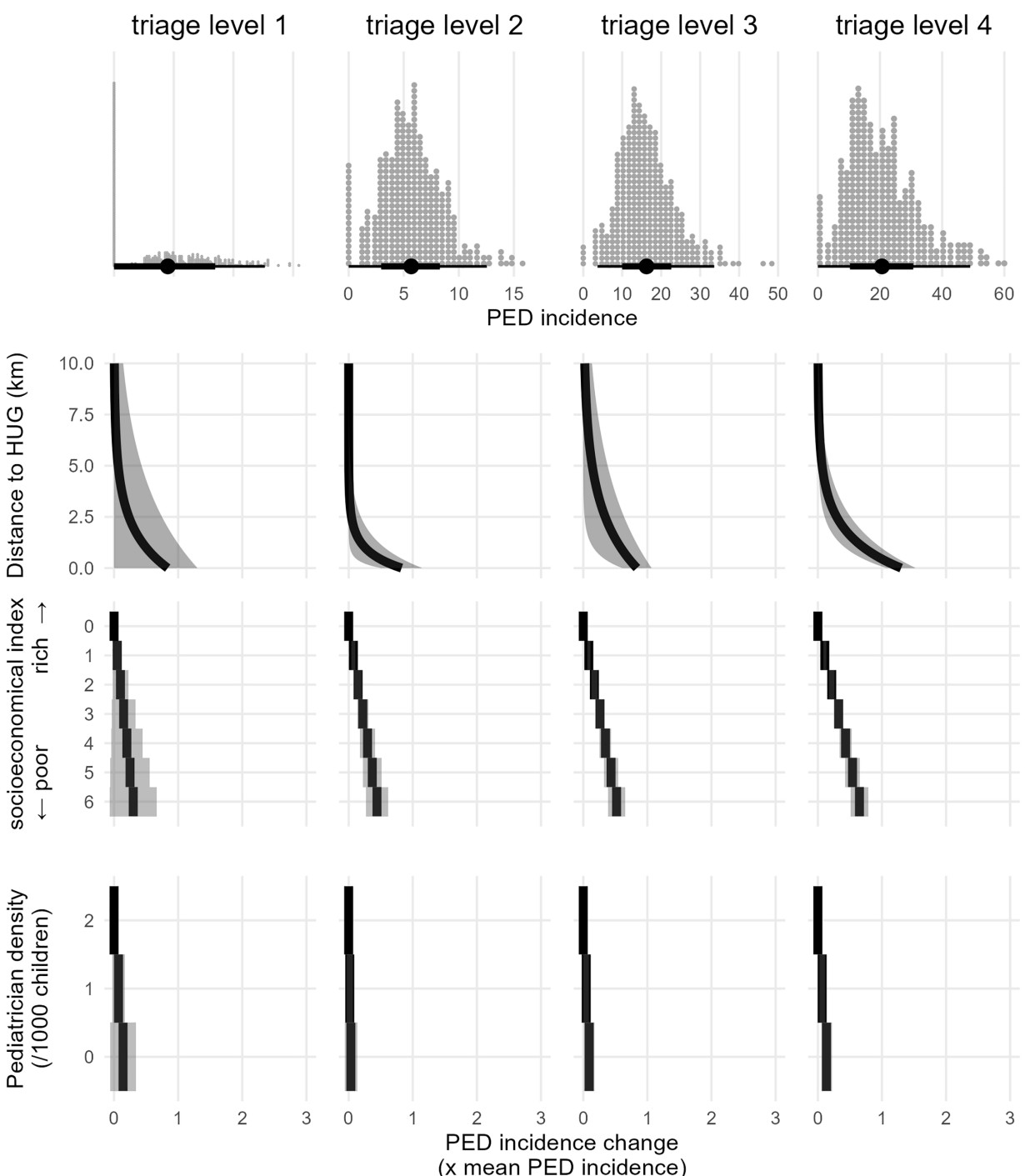

**Fig 2. Effects of key exposure variables on PED use, stratified by triage level (columns).** The top row shows the distribution of the neighborhood-level PED incidence. The three lower rows display variation in PED use incidence associated with (1) distance to the PED, (2) the neighborhood socio-economic index, and (3) pediatrician density. Effects are expressed relative to the mean incidence for each triage level. HUG, Geneva University Hospitals. PED, Pediatric Emergency Department.

**Table 4. Estimated number of PED visits during the 2023-2024 period, as predicted by the adjusted regression model under real-world conditions and four counterfactual scenarios: (1) all Geneva neighborhoods located 5 km from the PED, (2) all neighborhoods classified as socio-economically wealthy, (3) all neighborhoods with a pediatrician density of 2 per 1,000 children, and (4) all of the above conditions combined. Results are stratified by Canadian Triage Acuity Scale (CTAS) level (2: emergent, 3: urgent, 4*: less urgent (4) and non-urgent (5)) for cases where the regression model provide significant effects.**

| CTAS<br>Counterfactual scenarios | 2 | 3 | 4* | all |
|---|---|---|---|---|
| Full model: Number of predicted patients | 5291 | 15231 | 19479 | 40981 |
| Distance to PED = 5 km:<br>Number of patients (difference with full model: N, %) | 4720 (−571, −10.8%) | 13402 (−1829, −12%) | 15429 (−4050, −20.8%) | 34531 (−6450, −15.7%) |
| NSVI = 0 (wealthy):<br>Number of patients<br>(difference with full model: N, %) | 4555 (−736, −13.9%) | 12783 (−2448, −16.1%) | 15647 (−3832, −19.7%) | 33873 (−7108, −17.3%) |
| Pediatrician density = 2 per 1,000 children<br>Number of patients<br>(difference with full model: N, %) | | 14949 (−282, −1.9%) | 18963 (−516, −2.6%) | 40118 (−863, −2.1%) |
| All counterfactual combined<br>Number of patients<br>(difference with full model: N, %) | 3932 (−1344, −25.5%) | 10634 (−4557, −30%) | 11029 (−8365, −43.1%) | 26452 (−14386, −35.2%) |

vulnerable socio-economic profile reduced visits by 736 (−13.9%) for CTAS 2, 2,448 (−16.1%) for CTAS 3, and 3,832 (−19.7%) for CTAS 4*; and 3) applying a uniform pediatrician density of 2 per 1,000 children resulted in modest reductions of 37 (−0.7%) for CTAS 2, 282 (−1.9%) for CTAS 3, and 516 (−2.6%) patients for CTAS 4* (Table 4).

## 4. Discussion

### 4.1. Main findings and interpretation

This retrospective study revealed substantial variation in neighborhood-level PED use among children aged 0–16 across Geneva, a high-income city with universal health coverage and advanced medical infrastructure. High acuity visit patterns were relatively uniform across neighborhoods, suggesting that severe emergencies are less affected by contextual factors. In contrast, lower-acuity visits showed marked spatial variability, with proximity to the PED and poorer neighborhood socio-economic index being both associated with substantial increases in low-acuity visit incidence, with steep non-linear distance gradients and clear monotonic socio-economic gradients for CTAS 3 and 4. On the other hand, the pediatrician density contribution was minor compared with the other two determinants. Model fit improved with decreasing acuity (pseudo-$R^2$ from around 5% for CTAS 1 to 63% for CTAS 4*, with minimal residual spatial autocorrelation), indicating that the factors considered explained a significant proportion of the variability of low acuity neighborhood PED use between neighborhoods. In the fully adjusted models, distance to the PED and neighborhood socio-economic vulnerability were the dominant contextual predictors of neighborhood PED use, with effects increasing sharply at lower acuity levels. For CTAS 4, the maximum distance effect approached 130% of the mean incidence and the socio-economic gradient about 65%, whereas pediatrician density was only modestly and inversely associated with incidence and only for CTAS 3–4 (≤7% difference between lowest and highest densities). Counterfactual predictions showed that placing all neighbourhoods 5 km from the PED or setting the socio-economic index to zero (wealthy) would each reduce CTAS 4 visits by about 20%, while uniformly high pediatrician density (2 per 1,000 children) would reduce them by only 2%. Combining all favorable conditions yielded a 35% reduction in CTAS 2–4 visits, highlighting proximity and socio-economic vulnerability as the main levers for reducing non-urgent neighborhood PED use.

#### 4.1.1. Geographic proximity. Proximity was a key determinant of neighborhood-level PED use. Neighborhoods within 5 km of the hospital had higher PED use, especially for non-urgent conditions. This pattern reflects the well-documented

"distance decay effect", whereby even modest travel distances reduce the likelihood of seeking care for less urgent needs [4,11,15]. Previous studies have reported mixed findings, likely due to methodological differences, as many modeled distance linearly or with simple thresholds [15]. By using an exponential decay model, we captured the sharp, short-range effect of distance accurately, which is particularly relevant in a dense urban environment like Geneva. Although some studies have evidenced stronger distance effects in socio-economically disadvantaged areas [29], we observed no significant interaction between distance and socio-economic vulnerability, suggesting that the effect of distance was consistent across the social gradient. Our findings align with Ludwick et al. [30], who similarly reported distance decay in PED use, with incidence halving every 2–5 km depending on acuity—an effect we found most prominent for CTAS level 4*. This reinforces the notion that even in compact, well-connected cities, spatial barriers (e.g., crossing city or canton boundaries) can meaningfully influence care-seeking behavior, particularly for non-urgent cases.

### 4.1.2. Socio-economic vulnerability.

Socio-economic vulnerability was a strong, independent predictor of neighborhood-level PED use, with an effect size comparable to that of distance. Neighborhoods with higher composite vulnerability scores had significantly higher visit rates, particularly for lowest acuity cases. For CTAS levels 4 and 5, the incidence gap between the most and least vulnerable areas reached 13.2 percentage points (about 65% of the mean incidence) and accounted for up to 20% of total visits. This pattern is consistent with findings from other high-income countries, such as the United Kingdom and the United States, where lower income or more deprived communities disproportionately rely on emergency services for conditions that may not require hospital-level care [11,29–33]. Our findings are in line with a Swiss multicenter study in which families facing financial hardship had roughly 2–3 times higher odds of using PED services for non-urgent conditions [6]. This supports the view that neighborhood socio-economic context is a key driver of non-urgent pediatric emergency care use in high-income, universal-coverage systems. Potential mechanisms underlying this association are considered in Section 4.1.4.

### 4.1.3. Pediatrician density.

Primary care pediatrician density had a limited but measurable inverse association with neighborhood-level PED use, observed only for CTAS levels 3 and 4*. This suggests that greater outpatient capacity may help reduce moderate, non-urgent visits, though the effect was small. In our counterfactual models, uniformly high pediatrician density accounted for less than 2% of total PED visits, indicating a minor influence compared to proximity or socio-economic factors. This contrasts with previous studies where higher pediatrician supply appeared to reduce PED use in unadjusted analyses, but the effect weakened after adjusting for socio-economic context [34]. No association was observed in unadjusted models, but a modest inverse effect emerged after adjustment, likely reflecting Geneva's atypical pediatrician distribution, where higher provider densities are found in more socio-economically vulnerable neighborhoods, contrary to patterns in many other high-income settings [35,36]. Overall, our results suggest that geographic proximity to a pediatrician alone is insufficient to meaningfully curb non-urgent PED use.

### 4.1.4. Mechanistic pathways linking socio-economic vulnerability and PED use.

Several interrelated mechanisms are likely to underlie the strong association between neighborhood socio-economic vulnerability and low-acuity PED use observed in this study. At a structural level, socio-economically vulnerable areas are more likely to concentrate foreign families, experiencing material deprivation, unstable employment and constrained access to social support [37]. These conditions can increase both underlying morbidity [38] and practical barriers to engaging with primary care [39], for example through inflexible work schedules, limited childcare options or transport difficulties, and may make hospital-based care relatively more attractive for conditions that could be managed in the community [6,11,33]. In the Swiss multicenter study, economic precariousness, measured as difficulty paying household bills, more than doubled the odds of a low-acuity PED visit compared with financially secure families [6], supporting the view that low-acuity emergency visits are more strongly linked to financial insecurity than to parental education, employment status, immigrant background or social support alone. Similar constraints are likely captured by our composite neighbourhood socio-economic vulnerability index.

Behavioral and informational pathways probably reinforce these structural mechanisms. Systematic reviews indicate that parents from more disadvantaged backgrounds are more likely to perceive acute childhood illness as urgent, to

anticipate barriers to accessing community services, and to seek the perceived safety and diagnostic capacity of hospital emergency departments for relatively minor conditions [40,41]. Additionally, health literacy appears to be a key cross-cutting determinant. Low caregiver health literacy has been repeatedly associated with higher overall PED use and a greater proportion of non-urgent visits in both North American and European settings [4,42–44]. Other known contributors include language and cultural barriers [42], and lack of a consistent primary care physician [45,46]. The need for professional reassurance is also an important motivation for parents, particularly when symptoms are ambiguous or persistent [47]. For newly arrived or non-native families, who may face multiple informational and language barriers, the PED is often perceived as a trustworthy first point of contact when navigating uncertain or unfamiliar health concerns [42,47].

Within this framework, our finding that neighborhoods with higher socio-economic vulnerability have substantially higher rates of CTAS 4–5 visits is consistent with a pathway where financial strain, constrained time resources, lower system and health literacy, and lower trust and continuity in primary care interact to increase reliance on the PED for reassurance and rapid assessment rather than for strictly medical necessity.

**4.1.5. Comparison with other high-income healthcare systems.** Our results are concordant with evidence from other high-income settings where universal or near-universal coverage coexists with pronounced social gradients in PED utilization. A recent systematic review and meta-analysis of social determinants of health and PED outcomes reported that low-income, neighborhood deprivation and proximity to an ED are strong predictors of higher ED use across diverse health systems [11]. In Canada, AlSaeed et al. found higher rates of PED visits in dissemination areas characterised by high material deprivation and residential instability, even after adjustment for distance to hospital [33]. These findings mirror the pattern we observed in Geneva, with socio-economically vulnerable neighborhoods exhibiting a disproportionate burden of low-acuity attendance despite widespread formal access to primary care.

A recent systematic review synthesized studies from high- and middle-income countries evidenced that only half of the study found an association between time to an ED and increased non-urgent utilization [15]. Our exponential distance specification adds nuance to this literature by demonstrating that, in a compact urban environment, the effect of proximity is highly non-linear, with a steep increase in low-acuity visit rates within the first few kilometers of the PED and negligible effects beyond approximately 5 km. Collectively, these converging findings suggest that our results are likely to be relevant for other high-income cities with similar patterns of urban density and primary care organization.

## 4.2. Policy implications

The observation that non-urgent PED use is shaped predominantly by spatial proximity and socio-economic context, rather than pediatrician density alone, has important implications for intervention design. Systematic reviews of strategies to reduce inappropriate or repeated PED attendance have found only modest and short-lived effects of isolated interventions such as discharge education or case management, highlighting the need for multi-component approaches that address both access and underlying social needs [8,48]. Our results support prioritizing such interventions in neighborhoods that are both socio-economically vulnerable and located close to the PED, where potential impact on low-acuity visits is greatest.

Several policy directions emerge. First, interventions that strengthen caregiver health literacy and navigation skills through targeted anticipatory guidance, culturally adapted educational materials and community health worker programmes [49–51]. Second, co-locating social and primary care services in or near hospitals could mitigate financial and practical barriers that drive low-acuity use [52]. Third, telephone or digital triage systems in high-incidence areas may redirect a proportion of low-acuity presentations to primary care without compromising safety, although existing evidence indicates that such systems require careful design to avoid widening inequities [53–55] Finally, at the system level, our results align with recommendations to incorporate neighbourhood-based measures of deprivation and distance into planning of urgent care alternatives and extended-hours primary care, so that new services are preferentially located where both need and PED reliance are greatest [56–58]

### 4.3. Novel contributions and future research

This study contributes to the growing literature on determinants of pediatric emergency care in several ways. First, we quantify, within a single analytical framework, the independent and joint contributions of neighborhood socio-economic vulnerability, pediatrician density and finely modelled distance to the PED, while stratifying by triage acuity. To our knowledge, few previous studies have combined a composite socio-economic index with a non-linear distance function and provider density at high spatial resolution in a universal coverage context, and fewer still have done so specifically for children. Second, by demonstrating that socio-economic vulnerability and distance each explain a substantial fraction of low-acuity utilization, with no evidence of interaction, we show that these factors represent distinct axes of inequity that are not readily addressed by adjustments to primary care supply alone. Third, we provide citywide, acuity-specific population attributable fractions for these contextual determinants, offering concrete targets for local planning.

Future work should link individual clinical data, including chronic conditions and prior use of services, with neighborhood characteristics to better distinguish need-related from access-related drivers of PED use. Mixed-methods studies in high-incidence neighborhoods could examine how families balance convenience, perceived urgency, trust and past experiences when choosing between primary care and the PED, and how digital informational and triage tools, such as mobile apps [59], are used in these decisions. Significant spatial autocorrelation in the postal-code random intercepts found in our study suggest that unmeasured spatially structured factors may contribute to variation in low-acuity PED use beyond the determinants considered here. Future studies should consider larger scale measurements, such as differences in transportation infrastructure, cultural composition, or the presence of alternative healthcare facilities.

At system level, modelling strategies similar to those used here could be applied to multicenter, geospatially enriched national datasets that combine harmonized hospital records, public health data and neighborhood-level indicators. Such work would allow estimation of the incidence and economic burden of avoidable PED visits across regions and linguistic areas, provide comparative effect-size estimates for key socio-spatial determinants, and generate directly usable inputs for planning alternative urgent-care or extended-hours primary-care services.

### 4.4. Strengths and limitations

A major strength of this study is the use of a complete, two-year dataset from a public PED in a well-defined urban setting, combined with precise geospatial and sociodemographic data at the neighborhood level. The use of standardized triage levels enabled acuity-specific analyses, and modeling distance as an exponential function allowed for a nuanced characterization of its non-linear effect. The application of fully adjusted mixed-effect models further minimized potential confounding.

However, this study also has several limitations. First, we did not directly assess individual health needs or morbidity, which may partly explain higher PED use in more vulnerable neighborhoods. Although structural and access-related factors were central to our analysis, unmeasured health differences could contribute. Second, the ecological design does not allow inference at the individual level and introduces potential ecological bias. Third, the measure considered for the distance from neighborhood to PED is the straight-line distances from the neighborhood centroid, which may not be perfectly representative of the actual travel time and thus of the associated burden. Finally, the study focused on a single city, which may limit generalizability to other regions with different healthcare systems or population composition. Replication in multicenter studies is warranted to strengthen external validity.

## 5. Conclusion and perspectives

This study demonstrates that substantial disparities in PED use exist in a compact, high-income city with universal health coverage. Utilization patterns were shaped not only by need, but also by spatial proximity and neighborhood socio-economic context. Proximity alone accounted for up to 15% of neighbourhood level visits, particularly for low-acuity cases, while socio-economic vulnerability was associated with even larger disparities. In contrast, pediatrician density had only

a modest role. These findings indicate that addressing these disparities may require targeted, locally tailored strategies, such as community outreach in neighborhoods near hospitals, initiatives to build trust and engagement with primary care, and programs to improve health literacy among families.

## Supporting information

**S1 File. Supplementary material, tables and figures. Supplementary Table 1**: Composition of the neighbourhood socio-economic vulnerability index (NSVI); **Supplementary Fig 1**: Number of PED entries as a function of the hour of the day (top row), of the day of the year (middle row), and of the age of the children (bottom row); **Supplementary Fig 2**: spatial distribution in the state of Geneva (per neighbourhood) of the Number of children, distance to the Pediatric Emergency department, the density of pediatricians in a 2 km radius, and of the socioeconomical vulnerability index; **Supplementary table 2**: Results of the multivariable regression modeling the incidence of PED use for children between 0 and 5 years; **Supplementary Fig 3**: effect of the different exposure of interest on the incidence of PED use for children ages 0–5 years, stratified by triage level (columns). **Supplementary table 3**: Results of the multivariable regression modeling the incidence of PED use for children between 6 and 15 years; **Supplementary Fig 4**: effect of the different exposure of interest on the incidence of PED use for children ages 6–15 years, stratified by triage level (columns); **Supplementary table 4**: Results of the multivariable regression modeling the incidence of PED use between 7h and 18h; **Supplementary Fig 5**: effect of the different exposure of interest on the incidence of PED use during the week school hours (7h-18h); **Supplementary table 5**: Results of the multivariable regression modeling the incidence of PED use for children between 19h and 06h; **Supplementary Fig 6**: effect of the different exposure of interest on the incidence of PED use during the week outside school hours (19h-06h); **Supplementary table 6**: Results of the multivariable regression modeling the incidence of PED use for children for 2023–2024 weekends (Saturday and Sunday); **Supplementary Fig 7**: effect of the different exposure of interest on the incidence of PED use during the weekends.
(DOCX)

## Author contributions

**Conceptualization:** Denis Mongin, Hervé Spechbach, Joachim Marti, Frederic Ehrler, Johan N. Siebert.

**Data curation:** Denis Mongin.

**Formal analysis:** Denis Mongin.

**Methodology:** Denis Mongin, Joachim Marti.

**Software:** Denis Mongin.

**Visualization:** Denis Mongin.

**Writing – original draft:** Denis Mongin, Johan N. Siebert.

**Writing – review & editing:** Denis Mongin, Hervé Spechbach, Joachim Marti, Frederic Ehrler, Johan N. Siebert.

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
