## [Decision Letter · Decision Letter 0]

26 Nov 2025

PONE-D-25-47161Driving factors in pediatric emergency department use: an ecological retrospective studyPLOS ONE

Dear Dr. Mongin,

Thank you for submitting your manuscript to PLOS ONE. After careful consideration, we feel that it has merit but does not fully meet PLOS ONE’s publication criteria as it currently stands. Therefore, we invite you to submit a revised version of the manuscript that addresses the points raised during the review process.

We look forward to receiving your revised manuscript.

Kind regards,

Afagh Hassanzadeh Rad

Academic Editor

PLOS ONE

Journal Requirements:

4. In the online submission form, you indicated that data will be made available upon request, conditioned to the approval of a proposal with a signed data access agreement and to the approval of the ethical commission concerned.

5. We note that Figure 1, S2, in your submission contain [map/satellite] images which may be copyrighted. All PLOS content is published under the Creative Commons Attribution License (CC BY 4.0), which means that the manuscript, images, and Supporting Information files will be freely available online, and any third party is permitted to access, download, copy, distribute, and use these materials in any way, even commercially, with proper attribution. For these reasons, we cannot publish previously copyrighted maps or satellite images created using proprietary data, such as Google software (Google Maps, Street View, and Earth). For more information, see our copyright guidelines: http://journals.plos.org/plosone/s/licenses-and-copyright.

a. You may seek permission from the original copyright holder of Figures 1, S2, to publish the content specifically under the CC BY 4.0 license.

Reviewers' comments:

Reviewer's Responses to Questions

**Comments to the Author**

1. Is the manuscript technically sound, and do the data support the conclusions?

Reviewer #1: Partly

Reviewer #2: Yes

2. Has the statistical analysis been performed appropriately and rigorously? 

Reviewer #1: I Don't Know

Reviewer #2: Yes

3. Have the authors made all data underlying the findings in their manuscript fully available?

Reviewer #1: Yes

Reviewer #2: Yes

4. Is the manuscript presented in an intelligible fashion and written in standard English?

Reviewer #1: Yes

Reviewer #2: Yes

5. Review Comments to the Author

Reviewer #1: This manuscript represents a scientifically rigorous investigation, providing a clear and well-substantiated analysis of spatial variation in pediatric emergency department (PED) utilisation. The findings demonstrate that low-acuity visits are strongly modulated by socio-economic vulnerability and proximity to the hospital, whereas high-acuity visits remain largely unaffected by contextual factors. Notable strengths include the comprehensive two-year dataset, high-resolution geospatial and socio-demographic information, standardised CTAS triage, and the application of fully adjusted mixed-effects models. The limitations, including the retrospective ecological design, absence of individual-level health data, and restriction to a single urban centre, are appropriately acknowledged. To further enhance the manuscript, the authors should elaborate on the mechanistic pathways underlying the impact of socio-economic vulnerability, contextualise the findings relative to healthcare systems in other high-income settings, provide a summary table or figure of primary effects, clearly articulate policy implications, and explicitly delineate the novel contributions of this study in the context of existing literature

Reviewer #2: Authors must clearly state in Discussion and Conclusion that no individual-level inference can be drawn.

Rewrite interpretations to reflect neighborhood-level associations only.

Why was centroid distance preferred over population-weighted centroids or street-network distance?

If unique patients are used, explain whether a child with multiple visits counts once per CTAS category or once overall.

Clarify whether distance was measured “as the crow flies” or using travel time / road network.

Typographical errors:

“Halving diatnce” → “Halving distance”.

“Conterfactual” → “Counterfactual”.

“Geneva resident= 1 (%)” is unclear

Table 1 mixes visits and patients; columns unclear.

6. PLOS authors have the option to publish the peer review history of their article (what does this mean?). If published, this will include your full peer review and any attached files.

Reviewer #1: No

Reviewer #2: No

---

## [Decision Letter · Decision Letter 1]

2 Apr 2026

PONE-D-25-47161R1Driving factors in pediatric emergency department use: an ecological retrospective studyPLOS One

Dear Dr. Mongin,

Thank you for submitting your manuscript to PLOS ONE. After careful consideration, we feel that it has merit but does not fully meet PLOS ONE’s publication criteria as it currently stands. Therefore, we invite you to submit a revised version of the manuscript that addresses the points raised during the review process.

**The manuscript has been further evaluated by two reviewers, and their comments are available below.**

**Could you please carefully revise the manuscript to address all comments raised?**

We look forward to receiving your revised manuscript.

Kind regards,

Ilse Bloom

Staff Editor

PLOS One

**Journal Requirements:**

Reviewers' comments:

Reviewer's Responses to Questions

**Comments to the Author**

1. If the authors have adequately addressed your comments raised in a previous round of review and you feel that this manuscript is now acceptable for publication, you may indicate that here to bypass the “Comments to the Author” section, enter your conflict of interest statement in the “Confidential to Editor” section, and submit your "Accept" recommendation.

Reviewer #1: (No Response)

Reviewer #3: (No Response)

2. Is the manuscript technically sound, and do the data support the conclusions?

Reviewer #1: Yes

Reviewer #3: Yes

3. Has the statistical analysis been performed appropriately and rigorously?

Reviewer #1: I Don't Know

Reviewer #3: Yes

4. Have the authors made all data underlying the findings in their manuscript fully available?

Reviewer #1: Yes

Reviewer #3: Yes

5. Is the manuscript presented in an intelligible fashion and written in standard English?

Reviewer #1: Yes

Reviewer #3: Yes

6. Review Comments to the Author

**Reviewer #1:** Thank you for the opportunity to review this manuscript. The authors have adequately addressed the points raised in my previous comments, and I find the revisions satisfactory.

**Reviewer #3:** As the statistical reviewer I will focus on methods and reporting. note that this is the first time I see this paper. The methods are well described and appropriate. The key limitations, including ecological bias, are discussed. perhaps be more careful when you imply individual level inference at times.

Major

1) use an appropriate research checklist, e.g. STROBE probably in this case.

2) explain if all the data were complete for the analyses, are they likely are at this level.

Minor

1) How was pediatrician density operationalised

2) how was the NSVI index validated?

3) I'm not sure the "simulations" should be called that - they are model derived predictions under the same assumptions and "all else being equal". other than that, it is an appropriate approach.

4) perhaps also examine moran's I on the random effects as well, not only the residuals.

7. PLOS authors have the option to publish the peer review history of their article (what does this mean?). If published, this will include your full peer review and any attached files.

**Do you want your identity to be public for this peer review?** For information about this choice, including consent withdrawal, please see our Privacy Policy.

Reviewer #1: No

Reviewer #3: No

---

## [Author Response · Author response to Decision Letter 2]

10 Apr 2026

Reviewer #1: Thank you for the opportunity to review this manuscript. The authors have adequately addressed the points raised in my previous comments, and I find the revisions satisfactory.

Authors: we thank the reviewer for its positive feedback.

Reviewer #3: As the statistical reviewer I will focus on methods and reporting. note that this is the first time I see this paper. The methods are well described and appropriate. The key limitations, including ecological bias, are discussed. perhaps be more careful when you imply individual level inference at times.

Authors: We thank the reviewer for its positive feedback.

Major

1) use an appropriate research checklist, e.g. STROBE probably in this case.

Authors: We completed the STROBE checklist and uploaded it along the manuscript.

2) explain if all the data were complete for the analyses, are they likely are at this level.

Authors:

Data were complete, as they are aggregated data and administrative data.

We added to the result, section 3.2:

“No missing data were present in the aggregated data.”

Minor

1) How was pediatrician density operationalised

Authors: Thank you for raising this point. The operationalization of pediatrician density was described in the Methods section under 'Exposures of interest.' Specifically, we retrieved the addresses of all non-hospital pediatricians practicing in the Canton of Geneva from the Swiss Federal Register of Medical Professions [22]. Each address was then geocoded using the state geocoding API [20] to obtain precise geographic coordinates. Rather than computing a crude density per neighborhood, we adopted a spatial smoothing approach: for each neighborhood, we counted all pediatricians located within a fixed 2 km radius of the neighborhood centroid and expressed this as a density per 1,000 children. This radius-based approach is more representative of the actual availability of pediatricians from a family's perspective, as it captures the nearby supply of providers regardless of whether they happen to fall within the same administrative unit. We have reviewed and complemented this paragraph which now reads:

“Pediatrician density was derived from the Swiss Federal Register of Medical Professions [22] from which the addresses of all non-hospital pediatricians practicing in the Canton of Geneva were extracted. Each address was geocoded using the state geocoding API [20] to obtain precise geographic coordinates. For each neighborhood, the number of non-hospital pediatricians located within a fixed 2 km radius of the neighborhood centroid was counted and expressed per 1,000 children. This radius-based measure reflects the availability of pediatric primary care from a family's perspective, as it captures the nearby supply of providers irrespective of administrative boundaries.”

2) how was the NSVI index validated?

Authors: The NSVI has not undergone formal psychometric validation. It is a composite score developed by the “Centre d'analyse territorial des inégalités à Genève” (CATI-GE), a joint university–state institute dedicated to studying socio-economic inequalities in Geneva. The index is used operationally by cantonal authorities to allocate targeted funding to deprived neighborhoods, providing a degree of face and content validity. It has also been used in peer-reviewed research, including a study by our group on access to COVID-19 testing and outcomes, where it showed strong associations with both testing uptake and disease severity (Mongin et al., eClinicalMedicine 2022; doi:10.1016/j.eclinm.2022.101352). We have added a brief note in the Methods section acknowledging that the NSVI has not been formally validated but has demonstrated consistent associations with health outcomes in prior work:

“Socio-economic Status was assessed using the Neighborhood Socio-economic Vulnerability Index (NSVI), a composite score developed by the Centre d'analyse territorial des inégalités à Genève (CATI-GE), a joint university–state institute dedicated to studying socio-economic inequalities in Geneva [21], The NSVI ranges from 0 to 6 and is constructed from state-provided indicators including income, unemployment, and social benefits (full criteria are provided in Supplementary Table 1). Although the NSVI has not undergone formal psychometric validation, it is used operationally by cantonal authorities to guide the allocation of targeted funding to deprived neighborhoods, supporting its face and content validity. The index has also demonstrated consistent associations with health outcomes in prior peer-reviewed research, including a study conducted by our group in which higher NSVI scores were associated with reduced access to COVID-19 testing and worse disease outcomes [22].”

3) I'm not sure the "simulations" should be called that - they are model derived predictions under the same assumptions and "all else being equal". other than that, it is an appropriate approach.

Authors: We agree with the reviewer and have replaced ''counterfactual simulations' with 'counterfactual estimation’ throughout the manuscript to better reflect that these are derived from the fitted model under hypothetical conditions, holding all other factors constant.

4) perhaps also examine moran's I on the random effects as well, not only the residuals.

Authors: Thank you for this suggestion. We computed Moran's I on the estimated postal-code random intercepts for each CTAS level. Values were non-significant for CTAS 1 and CTAS 2, but significant for CTAS 3 (0.23, p<0.001) and CTAS 4* (0.16, p<0.05). This indicates that, for lower-acuity visits, some broad-scale spatial structure at the postal-code level remains unexplained by our exposure variables. This is not unexpected: while the exponential distance function captures the sharp, short-range proximity effect, broader geographic pattern, such as differences in urban fabric, transportation infrastructure, cultural composition, or the presence of alternative healthcare facilities, may vary smoothly across postal codes in ways not fully captured by our variable of interest. Importantly, Moran's I on the neighborhood-level residuals remained low and non-significant for all CTAS levels, confirming that the combination of fixed effects and postal-code random intercepts adequately accounts for spatial dependence at the analytical unit of interest. The spatial correlation in the random effects thus points to potential avenues for future research. These results have been added to the Results section:

“Residual spatial autocorrelation (Moran’s I) was below 0.1 and non-significant in all models, while spatial autocorrelation in the postal-code random intercepts was significant for CTAS 3 and 4* but below 0.3 (0.23 and 0.16 respectively).”

The corresponding values are reported in Table 2. We added in the discussion, in the paragraph about future work:

“Significant spatial autocorrelation in the postal-code random intercepts found in our study suggest that unmeasured spatially structured factors may contribute to variation in low-acuity PED use beyond the determinants considered here. Future studies should consider larger scale measurements, such as differences in transportation infrastructure, cultural composition, or the presence of alternative healthcare facilities.”

---

## [Decision Letter · Decision Letter 2]

23 Apr 2026

Driving factors in pediatric emergency department use: an ecological retrospective study

PONE-D-25-47161R2

Dear Dr. Denis Mongin,

We’re pleased to inform you that your manuscript has been judged scientifically suitable for publication and will be formally accepted for publication once it meets all outstanding technical requirements.

Kind regards,

Helga Naburi, MD, Mmed,MPH,PhD

Academic Editor

PLOS One

Additional Editor Comments (optional):

Reviewers' comments:

Reviewer's Responses to Questions

**Comments to the Author**

1. If the authors have adequately addressed your comments raised in a previous round of review and you feel that this manuscript is now acceptable for publication, you may indicate that here to bypass the “Comments to the Author” section, enter your conflict of interest statement in the “Confidential to Editor” section, and submit your "Accept" recommendation.

Reviewer #3: All comments have been addressed

2. Is the manuscript technically sound, and do the data support the conclusions?

Reviewer #3: Yes

3. Has the statistical analysis been performed appropriately and rigorously? 

Reviewer #3: Yes

4. Have the authors made all data underlying the findings in their manuscript fully available?

Reviewer #3: Yes

5. Is the manuscript presented in an intelligible fashion and written in standard English?

Reviewer #3: Yes

6. Review Comments to the Author

Reviewer #3: I am satisfied with the responses and the resulting changes to the paper. I have nothing else to add.

7. PLOS authors have the option to publish the peer review history of their article (what does this mean?). If published, this will include your full peer review and any attached files.

Reviewer #3: No

---

## [Editor Report · Acceptance letter]

PONE-D-25-47161R2

PLOS One

Dear Dr. Mongin,

I'm pleased to inform you that your manuscript has been deemed suitable for publication in PLOS One. Congratulations! Your manuscript is now being handed over to our production team.

Kind regards,

on behalf of

Dr. Helga Naburi

Academic Editor

PLOS One